# BR-Mediated Protein *S*-Nitrosylation Alleviated Low-Temperature Stress in Mini Chinese Cabbage (*Brassica rapa* ssp. *pekinensis*)

**DOI:** 10.3390/ijms231810964

**Published:** 2022-09-19

**Authors:** Xueqin Gao, Jizhong Ma, Jianzhong Tie, Yutong Li, Linli Hu, Jihua Yu

**Affiliations:** 1College of Horticulture, Gansu Agricultural University, Lanzhou 730070, China; 2Gansu Provincial Key Laboratory of Aridland Crop Science, Gansu Agricultural University, Lanzhou 730070, China

**Keywords:** brassinosteroids, nitric oxide, *S*-nitrosylation, low-temperature stress, *Brassica rapa* ssp. *pekinensis*

## Abstract

Brassinosteroids (BRs), a novel plant hormone, are widely involved in plant growth and stress response processes. Nitric oxide (NO), as an important gas signaling molecule, can regulate target protein activity, subcellular localization and function in response to various stresses through post-translational *S*-nitrosylation modifications. However, the relationship between BR and NO in alleviating low-temperature stress of mini Chinese cabbage remains unclear. The hydroponic experiment combined with the pharmacological and molecular biological method was conducted to study the alleviating mechanism of BR at low temperature in mini Chinese cabbage. The results showed that low temperature inhibited the growth of mini Chinese cabbage seedlings, as evidenced by dwarf plants and yellow leaves. Treatment with 0.05 mg/L BR and 50 µM NO donor S-nitrosoglutathione (GSNO) significantly increased the leaf area, stem diameter, chlorophyll content, dry and fresh weight and proline content. Meanwhile, the malondialdehyde (MDA) content in 0.05 mg/L BR- and 50 µM GSNO-treated leaves were significantly lower than those in other treated leaves under low-temperature conditions. In addition, BR and GSNO applications induced an increase in NO and S-nitrosothiol (SNO) levels in vivo under low-temperature stress. Similarly, spraying BR after the elimination of NO also increased the level of *S*-nitrosylation in vivo, while spraying GSNO after inhibiting BR biosynthesis decreased the level of NO and SNO in vivo. In contrast, the S-nitrosoglutathione reductase (*BrGSNOR*) relative expression level and GSNOR enzyme activity were downregulated and inhibited by BR treatment, GSNO treatment and spraying BR after NO clearance, while the relative expression level of *BrGSNOR* was upregulated and GSNOR enzyme activity was also increased when spraying GSNO after inhibiting BR synthesis. Meanwhile, the biotin switch assay showed that exogenous BR increased the level of total nitrosylated protein in vivo under low-temperature stress. These results suggested that BR might act as an upstream signal of NO, induced the increase of NO content in vivo and then induced the protein *S*-nitrosylation modification to alleviate the damage of mini Chinese cabbage seedlings under low-temperature stress.

## 1. Introduction

Brassinosteroids (BRs) are a substance with high biological activity. They were first isolated and extracted from rape pollen by Mitchell et al. [1]. In 1979, Grove et al. identified brassinosteroid as a sterol compound, which was officially named brassinolactone [2]. As a new plant hormone, up to now, more than 70 BR-related compounds have been identified, among which 2,4-epbrassinolide (2,4-EBR) and 2,8-high brassinolide (2,8-HBR) are the two most active, which have been widely used in the study of BR function and signal transduction [3]. Growth biomarkers, SPAD chlorophyll and net photosynthetic rate were significantly reduced in ‘S-22’ (cold-tolerance variety) and ‘PKM-1’ (cold-sensitivity variety) grown at 20 °C/14 °C, 12 °C/7 °C and 10 °C/3 °C. However, 2,4-EBR combined with hydrogen peroxide (H_2_O_2_) significantly improved the above parameters and increased the activities of different antioxidant enzymes (CAT and SOD) and proline contents. Thus, the tolerance of tomatoes to low temperature was enhanced [4]. Related studies showed that exogenous EBR treatment increased the contents of gibberellin (GA3) and indole-3-acetic acid (IAA), reduced the damage of reactive oxygen species (ROS) and promoted the growth rate of tomatoes under low-temperature stress [5]. In the presence of BRs, the kinase activity of BRI1 is activated, and BRI1 interacts with its co-receptor BAK1 (BRI1-associated receptor kinase 1) to form a complex. BRI1 and BAK1 phosphorylate each other and release the BRI1 inhibitor BKI1 (BRI1 kinase inhibitor 1) at the same time. Then, the phosphate cascade process is completed through BSK1 (BR-signaling kinase 1), CDG1 (constitutive differential growth 1) and other signaling elements, which activates BSU1 (BRI1-suppressor 1) to inhibit BIN2 and dephosphorylates BES1/BZR1 [6]. Interestingly, BZR1 (Brassinazole resistant 1), an important transcription factor for BR signaling, can bind directly to the promoter regions of *CBF1* and *CBF2* (CBF/DREB) and promote their expression in response to low-temperature stress, further illustrating the promotion of BR signaling for low-temperature resistance [7]. Meanwhile, BR induces BZR1 to directly activate the expression of the RBOH1 (respiratory burst oxidase homolog 1) and the accumulation of apoplastic H_2_O_2_, thereby increasing the protein involved in photoprotection, suggesting that this process may be related to the autophagic pathway [8]. In *Solanum Lycopersicum*, the overexpression *Dwarf* (BR biosynthetic gene) results in increased carotenoid biosynthesis with low-temperature stress, and BR positively regulates the photoprotection process. Overexpression of the BR synthesis gene *DWARF4* induces accumulation of *COR15A* (cold regulate, COR) transcripts, thereby enhancing cold resistance in *Arabidopsis* [9]. In addition, it was found that EBR treatment could reduce the effect of low-temperature stress on maize growth by increasing plant height, chlorophyll and soluble protein content [10]. Therefore, exogenous BR plays an important regulatory role in alleviating low-temperature stress in plants, but the mechanism of how BR alleviates low-temperature stress in mini Chinese cabbage is still unclear.

Protein sulfhydryl nitrosylation occurs when NO modifies the cysteine (Cys) sulfhydryl (-SH) of the target protein to produce *S*-nitrosothiols (SNO), to realize the regulation process of redox signal transduction [11,12]. As an important post-translational modification, protein *S*-nitrosylation plays a very important role in regulating abiotic stresses [13]. It has been reported that of 61 proteins regulated by *S*-nitrosation in sunflower seedlings under salt stress, 17 were specific to cotyledons, 4 were specific to roots and 40 were common to both, among which the most *S*-nitrosylated proteins were carbohydrate metabolism, followed by other metabolic proteins. These results suggested that NO could regulate the energy balance of various metabolites in sunflower seedlings under salt stress [14]. APX was regulated by *S*-nitrosylation in peas under salt stress, and *S*-nitrosylation in Cys32 (cysteine 32) significantly increased APX activity, suggesting a link between *S*-nitrosylation and related antioxidant enzymes involved in ROS metabolism [15]. In addition, *S*-nitrosylation of APX1 on Cys-32 enhanced the H_2_O_2_ scavenging activity of APX1, thus enhancing the antioxidant capacity of *Arabidopsis thaliana*. However, the Cys-32 mutation resulted in reduced APX1 activity [16]. These results suggested that *S*-nitrosylation of APX plays an important role in the response mechanism of nitrogen oxidation. Bai et al. [17] reported that protein *S*-nitrosylation levels of antioxidant enzymes were significantly reduced in seeds under dehydration. On the contrary, the addition of NO increased seed tolerance to drought by enhancing *S*-nitrification. Abat et al. [18] reported that 17 different *S*-nitrosylation sites were identified in low-temperature-stressed *Brassica juncea*. In addition, the content of SNO under low-temperature stress was significantly increased by 1.4 times at 1 h. Low temperature affects the photosynthesis of plants and reduces their photosynthetic efficiency of plants [19]. It was found that the activity of GSNOR was significantly increased in peas induced by low temperature. At the same time, the contents of NO and SNO also increased, which caused *S*-nitrosylation in pea to relieve the low-temperature stress [20]. Moreover, forty-eight *S*-nitrosylated proteins were identified in the extraneous germs of mustard seedlings. *S*-nitrosylation increased the activities of dehydroascorbate reductase (DHAR) and glutathione *S*-transferase (GST) to remove reactive oxygen species, thereby enhancing cold tolerance [21]. However, SNO levels and GSNOR activity decreased in *Arabidopsis thaliana* seeds treated with 150 µM Cd for 14 d [22]. Perazzolli et al. [23] found that anaerobic treatment of *Arabidopsis* leaves for 24 h reduced SNO levels and GSNOR activity in vivo. In conclusion, when plants are under abiotic stresses, the level of in vivo *S*-nitrosylation will change, thus affecting the activity of target proteins in stresses, and ultimately regulating the response of plants to abiotic stresses.

Low temperature (LT) is one of the main factors restricting plant growth and development, directly affecting crop yield and quality. When serious, it can cause irreversible damage or even death of plants and cause serious losses to agricultural production. The damage caused by low temperature to plants can be divided into chilling stress (0~20 °C) and freezing damage (<0 °C) [24], which mainly affect the normal growth and development of plants in three ways: destroying the plant cell membrane system, affecting plant photosynthesis and changing plant osmotic regulation [19]. Mini Chinese cabbage (*Brassica rapa* ssp. *pekinensis*), belonging to the subspecies of *Brassica* in Cruciferae, becomes the main cultivar of summer vegetables on a plateau of its high economic efficiency [25]. However, in the spring sowing season in the high cold and cool areas, Chinese cabbage often suffers from low temperatures, such as reverse spring cold. As a result, Chinese cabbage quickly completed vernalization and the phenomenon of “early bolting flowering” appears [26], which seriously affects its yield and quality. Ding et al. [27] showed that in cold-stressed cucumber, NO content increased after 24-epibrassinolide (EBR) treatment and decreased after inhibiting BR biosynthesis. Moreover, young alfalfa (*Medicago truncatula*) leaves treated at 4 °C were sprayed with BR and BRz (BR biosynthesis inhibitor), which a large amount of NO was accumulated under BR treatment. However, when BR biosynthesis was inhibited, the content of NO decreased significantly and reduced cold resistance [28]. So far, studies on the role of BR in plant response to abiotic stresses have been increasing gradually, but few reports have focused on BR involvement in NO-induced *S*-nitrosylation to alleviate the low-temperature stress of mini Chinese cabbage. To explore whether exogenous BR alleviates low-temperature stress in mini Chinese cabbage seedlings by increasing NO content and enhancing the *S*-nitrosylation level of proteins, in this study, we first cleared that exogenous BR and GSNO at optimum concentration could alleviate low-temperature stress in mini Chinese cabbage, and the upstream or downstream relationship of BR and NO was discussed, finally, the protein *S*-nitrosylation levels induced by BR and GSNO under low-temperature stress were measured. Our results preliminarily demonstrated that BR acts as an upstream signal of NO and induced protein *S*-nitrosylation, which was involved in alleviating low-temperature stress in mini Chinese cabbage seedlings. The specific *S*-nitrosylation proteins induced by BR under low temperature would be identified in the following experiment, to further elaborate the mechanism of BR-mediated protein *S*-nitrosylation modification to improve the low-temperature tolerance of mini-Chinese cabbage, which provided a new strategy for the management of mini Chinese cabbage under low temperature and a new research idea and direction for the mitigation mechanism of BR on low-temperature stress.

## 2. Results

### 2.1. Screening of the Optimum Concentration of BR for Alleviating Low-Temperature Stress of Mini Chinese Cabbage Seedlings

To explore the optimal concentration of BR to alleviate low-temperature stress, seedlings were treated with different concentrations of BR (0, 0.01, 0.05, 0.1, 0.5 and 1 mg/L) in the experiment. As shown in Figure 1A, low temperature significantly inhibited the growth of seedlings, making their leaves yellow. Among them, the 0.05 mg/L BR treatment had the most effective alleviating effect. As can be seen from Figure 1B–E, low temperature significantly reduced the leaf area, chlorophyll contents, proline contents and increased MDA contents. Compared with low-temperature treatment, spraying BR on leaves had different effects on the indexes. The leaf area and chlorophyll content showed a trend of increasing and then decreasing with the increase of BR concentration. The leaf area under the 0.01 mg/L BR and 1.0 mg/L BR treatments was the smallest, while the leaf area, chlorophyll contents and proline contents under the 0.05 mg/L BR treatment were significantly higher than those under low-temperature stress, increased by 75.6%, 15.6% and 246.5%, respectively. Subjected to the 0.05 mg/L BR treatment, the MDA contents decreased by 12.9% compared with LT. Combined with phenotype, growth and physiological indexes, 0.05 mg/L BR was selected as the optimal concentration to alleviate low-temperature stress of mini Chinese cabbage seedlings.

### 2.2. Screening of the Optimum Concentration of GSNO for Alleviating Low-Temperature Stress of Mini Chinese Cabbage Seedlings

To explore the optimal concentration of GSNO (NO donor) to alleviate low-temperature stress, seedlings were treated with different concentrations of GSNO (0, 10, 50 and 150 μM) in the experiment. As shown in Figure 2A, low temperature significantly inhibited the growth of mini Chinese cabbage seedlings and reduced their leaf area, while exogenous application of 50 μM GSNO effectively alleviated the damage caused by low-temperature stress and significantly improved the cold tolerance of mini Chinese cabbage seedlings. In addition, low temperature significantly reduced the leaf area, chlorophyll, and proline and increased MDA. However, the low concentration of GSNO (50 μM) increased leaf area, chlorophyll content and free proline content by 1.25-, 1.10- and 7.19-fold, respectively (Figure 2B–E). Meanwhile, high concentrations of GSNO (100 μM, 150 μM) significantly inhibited the leaf area, chlorophyll and proline content of mini Chinese cabbage. The MDA content was significantly lower under 50 μM GSNO treatment compared with other low-temperature stress treatments. These results suggested that exogenous GSNO alleviates low-temperature stress in mini Chinese cabbage seedlings with a concentration-dependent effect. Finally, 50 μM GSNO was chosen as the optimum concentration spraying on leaves to improve the low-temperature tolerance of mini Chinese cabbage seedlings.

### 2.3. Effects of BR, GSNO, BRz and cPTIO on the Growth of Mini Chinese Cabbage Seedlings under Low-Temperature Stress

Based on the results of the above concentration screening experiments, subsequent experiments were conducted using 0.05 mg/L BR and 50 μM NO. To explore the effects of BR, GSNO, BRz and cPTIO on the growth of mini Chinese cabbage seedlings under low-temperature stress, we treated mini Chinese cabbage seedlings with BR, GSNO, BRz (BR biosynthesis inhibitor) and cPTIO (NO scavenger). As shown in Figure 3, low temperature significantly inhibited seedling growth, mainly by reducing dry weight, fresh weight, stem diameter and leaf area. Exogenous spraying of BR and GSNO effectively alleviated the low-temperature stress and significantly increased dry weight, fresh weight, stem diameter and leaf area. However, the addition of cPTIO and BRz reduced the dry weight as well as leaf area, and there was no significant difference between them. These results indicated that exogenous spraying of 0.05 mg/L BR and 50 μM NO improved the cold tolerance of mini Chinese cabbage seedlings.

### 2.4. Effects of BR, GSNO, BRz and cPTIO on Endogenous NO and SNO Levels under Low-Temperature Stress

To explore the upstream or downstream relationship between BR and NO in the process of alleviating low-temperature stress by regulating *S*-nitrosylation in vivo, the endogenous NO and SNO levels were measured. The changes in NO content in the leaves of mini Chinese cabbage seedlings under different treatments were shown in Figure 4A. Low temperature significantly decreased NO content compared with normal temperature. However, foliar spraying of BR and GSNO significantly increased the level of endogenous NO, which was 1.28-fold and 1.21-fold higher than LT treatment, respectively. Interestingly, we found that after removing NO and then foliar spraying BR, the level of endogenous NO increased and was not significantly different from the BR and GSNO treatments. However, inhibition of the biosynthetic pathway of BR in vivo followed by the addition of GSNO significantly reduced the level of endogenous NO and was not significantly different from the LT treatment.

To further demonstrate the upstream or downstream relationship between BR and NO, the changes of SNO contents in leaves were measured (Figure 4B). Under the CK treatment, the SNO content was significantly higher than in the other treatments. Meanwhile, BR treatment and GSNO treatment significantly increased the level of endogenous SNO, which were 1.28-fold and 1.32-fold higher than LT treatment, respectively. Notably, we found that foliar spraying of BR after removing NO in vivo increased the level of endogenous SNO and was not significantly different from the BR and NO treatments. However, after inhibiting the synthesis of BR in vivo, spraying GSNO significantly reduced the level of endogenous SNO and was not significantly different from the LT treatment. These results suggested that exogenous BR might act as an upstream signal of NO to induce the increase of NO and SNO levels in vivo, thereby alleviating the low temperature stress of mini Chinese cabbage.

### 2.5. Effects of BR, GSNO, BRz and cPTIO on BrGSNOR Gene Expression and GSNOR Enzyme Activity under Low-Temperature Stress

Compared with CK, the relative expression of the *BrGSNOR* gene was upregulated at low temperature. However, foliar spraying of BR and NO had a significant effect on the expression of the *BrGSNOR* gene in mini Chinese cabbage. As shown in Figure 5A, the expression of *BrGSNOR* genes was downregulated in BR treatment and GSNO treatment, both of which were significantly lower than in LT treatments. Similarly, the relative expression of the *BrGSNOR* gene was down-regulated when spraying BR after removing NO. On the contrary, when the biosynthesis of BR was inhibited, the *BrGSNOR* gene was upregulated by exogenous spraying of GSNO and was not significantly different from the LT treatment. GSNOR enzyme is the key index to maintain the balance of nitrosylation in vivo. To further verify whether the changing trend of GSNOR enzyme activity is consistent with the relative expression of the *BrGSNOR* gene, GSNOR enzyme activity was measured in this experiment. The GSNOR activity under different treatments was shown in Figure 5B, which was consistent with the trend of *BrGSNOR* gene expression. LT + BR, LT + GSNO and LT + BR + cPTIO treatments significantly inhibited the GSNOR activity, and there was no significant difference among the three treatments. LT and LT + GSNO + BRz treatments significantly increased the activity of GSNOR and there was no significant difference between the two treatments. The above results were in contrast to the NO and SNO levels in vivo, and these results implied that, under low-temperature stress, exogenous BR acts as an upstream signal for NO, differentially regulating the expression of *BrGSNOR* genes as well as their enzymatic activity to maintain the balance of *S*-nitrosylation in vivo.

### 2.6. Analysis of Total Nitrosylated Proteins under Different Treatments

In this experiment, the biotin switch assay was used to further determine whether there is a possible modification of protein nitrosylation induced by exogenous BR and GSNO in improving the low-temperature tolerance of mini Chinese cabbage seedlings. As shown in Figure 6, there were clear bands around 60 KDa and 45 KDa in all six treatments. However, the bands around 100 KDa were very shallow. The darkest and widest protein band was observed under the LT + BR treatment, suggesting that the highest number of nitrosylated modified proteins was observed under this treatment, followed by the CK treatment. The protein bands under low temperature treatment were significantly narrower than those under CK and BR treatment. The least amount of nitrosylated modified proteins occurred under the LT + GSNO + BRz treatment. The bands under the LT + NO and LT + BR + cPTIO treatments were narrower and the color of the bands was darker than that of LT + NO + BRz. Similarly, the results of biotin bands were consistent with the relative gray value in Figure 6B. It was concluded that scavenging of NO in vivo and exogenous spraying of BR induced the occurrence of *S*-nitrosylation modification while inhibiting the biosynthetic pathway of endogenous BR and exogenous spraying of GSNO attenuated the extent of protein *S*-nitrosylation modification.

## 3. Discussion

Cold and frost damage are common environmental factors which many crops face in northwest China, severely limiting crop production [29]. Low temperatures lead to reduced seed germination, significant inhibition of plant growth and reproduction and substantial reduction in crop yield and quality [30,31]. Early studies have shown that BR played an important function in cold stress [32,33]. Exogenous BRs enhanced the expression of *CBF1* and *COR47* after low-temperature treatment, indicating that BRs enhanced the expression and low-temperature resistance of *Arabidopsis CBF* [34]. In addition, NO is an active nitrogen species that can directly participate in plant response to abiotic stress through *S*-nitrosation modification. *S*-nitrosation of the transcription factor *LeCBF1* was found in tomato seedlings treated with cold damage [35]. In this study, low temperature inhibited the growth and chlorophyll synthesis of the seedlings, resulting in yellow leaves. At the same time, the content of MDA increases, aggravating the degree of membrane lipid peroxidation, thus damaging the integrity of cell membrane. The content of free proline is also affected, which is reduced by low temperature and leads to cell water loss. In contrast, exogenous spraying of 0.05 mg/L BR and 50 µM GSNO effectively alleviated the low-temperature damage in mini Chinese cabbage. Therefore, this study investigated the alleviating effects of exogenous BR and GSNO on low temperature and the upstream or downstream relationship between BR and NO. From the perspective of *S*-nitrosylation modification induced by BR, the mechanism of alleviating low temperature stress by BR was preliminarily clarified.

### 3.1. The Alleviation Role of BR on Low Temperature in Mini Chinese Cabbage Seedlings

In this experiment, exogenous BR significantly enhanced the growth of mini Chinese cabbage seedlings under low-temperature stress and reduced the harmful effects of low temperature (Figure 1). These results confirm the results of earlier studies [8,36,37]. Chlorophyll is the most important pigment in photosynthesis. Low-temperature stress will destroy the selective permeability, fluidity and membrane composition changes of the chloroplast thylakoid membrane, and then affect the content of chlorophyll [38]. In this study, it was found that spraying BR could significantly increase chlorophyll content (Figure 1C). The effect was significant at 0.05 mg/L of BR concentration, but not at a higher or lower concentration, which was consistent with the results of a previous study on pepper and cucumber [39,40], which revealed that BR has a concentration effect on plant growth. Studies have shown that when plants are under stress, the balance between the production and clearance of free radicals in cells is destroyed, leading to the accumulation of free radicals such as OH^−^ and O_2_^−^, which destroys the integrity of cell membranes and leads to the accumulation of MDA, causing damage to plants. Exogenous BR can reduce H_2_O_2_ content and O_2_^−^ production rate and MDA content under stresses [41,42]. The results showed that the MDA content in the seedlings under low-temperature stress was significantly higher than that of the control. Spraying 0.05 mg/L BR significantly reduced the MDA content in the leaves, effectively reduced the degree of membrane lipid peroxidation and thus enhanced the ability of plants to resist low-temperature stress. Under stresses, plants evolved to form some macromolecules, which play an important role in maintaining an osmotic balance of cell fluid [43]. The results of this study showed that low temperature increased the content of proline and spraying BR at low temperature produced a higher level of proline, suggesting that BR may enhance the tolerance of plants to low temperature by increasing soluble organic matter. The BR-induced accumulation of organic and inorganic substances is involved in plant response to stress and normal cellular metabolism [44]. Our data suggested that the low-temperature stress of mini Chinese cabbage was effectively alleviated by 0.05 mg/L BR application to some extent.

### 3.2. The Alleviation Role of GSNO on Low Temperature in Mini Chinese Cabbage Seedlings

When plants suffer from drought, low temperature and other stresses, various mechanisms are formed in plants to cope with and resist stress, such as scavenging reactive oxygen species and producing stress-resistant proteins or other metabolites. As one of the four major gas signaling molecules, NO is an important stress substance and plays an important role in the oxidative defense of plant cells [45,46]. Photosynthetic pigments participate in the absorption, transfer and transformation of plant light energy and are an important indicator of photosynthetic capacity. To a large extent, plants have strong self-adaptation and adjustment ability to a light environment. Some plants may capture more light energy by synthesizing a large amount of chlorophyll under low light, to adapt to the bad environment [47]. It was found that in leaves of *Arabidopsis* mutant with NO deficiency type *nos1*/*noa1*, genes for key enzymes of chlorophyll degradation were significantly upregulated at the transcriptional level, while the wild type could effectively preserve chlorophyll content during leaf senescence and delay leaf senescence [48]. It is worth noting that in this experiment, 50 µM GSNO treatment of mini Chinese cabbage seedlings under low-temperature stress can effectively improve the content of chlorophyll in leaves. In addition, when leaf tissues were subjected to low temperature stress, the degree of oxidative stress deepened, increasing MDA content. However, exogenous NO treatment reduced the MDA content, reduced the products of lipid peroxidation of the cell membrane and effectively reduced the permeability of the cell membrane. This is similar to the research results of Yong et al. [49] and Xu et al. [50]. Proline is an important substance widely accumulated to regulate the REDOX state of plants under stress. Studies have shown that proline mostly exists in plant cytoplasm, which can not only regulate the balance between cytoplasm and vacuoles but also acts as an important antioxidant, which can cooperate with the antioxidant system to remove hydroxyl free radicals, superoxide anion free radicals and other substances in cells [51,52]. Previously, increased NO in the cold has been shown to actively regulate the accumulation of proline, an amino acid-based osmotic pressure. These results indicated that NO signal is involved in osmotic maintenance by regulating osmotic factors. Consistent with this, in this study, exogenous spraying of 50 µM GSNO effectively increased the content of intracellular proline, thus reducing the oxidative damage of vegetables caused by low temperature stress, indicating that NO plays a certain regulatory and protective role in membrane lipid peroxidation. The above results indicated that exogenous spraying of 50 µM GSNO can effectively improve the tolerance to low-temperature stress.

### 3.3. BR as NO Upstream Signal Induced Protein S-Nitrosylation Alleviated Low-Temperature Stress in Mini Chinese Cabbage Seedlings

NO plays its biological functions by substituting hydrogen ions of sulfhydryl groups in specific cysteine residues in proteins to form covalently linked nitrosothiols (S-NO) [53], that is, protein post-translational modification *S*-nitrosylation. As an important post-translational modification, it has attracted extensive attention in recent years [13]. Previously, Xiong et al. [54] studied the effect of Cd stress on endogenous NO content in rice seedlings. The results showed that Cd significantly reduced the endogenous NO level of rice seedlings by inhibiting NOS activity. However, the inhibitory effect of Cd stress could be alleviated by the exogenous application of NO donor sodium nitropropyl (SNP). Similar results were found in *Medicago* [55]. In this experiment, the mechanism of BR alleviating low-temperature stress by regulating the level of nitrosation in mini Chinese cabbage seedlings was explored. It was found that low-temperature stress reduced NO content in mini Chinese cabbage seedlings. However, spraying exogenous BR could increase the NO content, and likewise, after removing endogenous NO and then spraying BR, could also increase the NO content, and there was no significant difference between the two treatments. We hypothesized that BR might act as an upstream signal of NO and induce endogenous NO production under low-temperature stress. Consistent with our results, a recent study by Aying et al. [56] found that BR can promote the antioxidant ability of maize treated with PEG, and ABA plays a role in BR promoting the antioxidant ability, and BR can upregulate the expression of a key gene of ABA synthesis, *vpl4*. BR can induce the production of NO, and NO can promote the synthesis of ABA, which is an important mechanism to enhance the drought resistance of maize leaves. Relevant studies have shown that adding exogenous substances, such as CaCl_2_ and GSNO, can significantly improve the SNO level in cucumbers under Cd stress [57]. To further investigate the potential of BR regulating plant physiological processes by mediating modified protein cysteine residues, in this study, the SNO level was determined in the process of improving the tolerance to low temperature. We found that the application of exogenous BR and GSNO significantly increased the endogenous SNO level and showed no significant difference with LT + BR + cPTIO. However, the SNO level under LT + GSNO + BRz treatment was significantly lower than that under BR and NO treatments (Figure 4B). Following the above research results, BR can induce the production of NO in mini Chinese cabbage under low-temperature stress, and, similarly, the level of SNO can also be increased through BR induction. In addition, GSNOR can regulate the level of *S*-nitrosylation in plants [58]. For example, Lin et al. found that overexpressed *OsGSNOR* had lower SNO content in H_2_O_2_-induced cell death in rice leaves, suggesting that GSNOR may play an important role in SNO homeostasis [59]. Exogenous NO significantly reduced the activity of the GSNOR enzyme during adventitious root formation [57]. In the present study, the *BrGSNOR* gene relative expression level was downregulated in the LT + BR, LT + NO and LT + BR + cPTIO treatments (Figure 5A), and the activity showed the same trend (Figure 5B), which also indicated that GSNOR regulated the total level of SNOs during BR-mediated *S*-nitrosylation. Notably, the relative expression of *BrGSNOR* was upregulated, and its enzyme activity was also significantly increased, while the SNO level was decreased by low-temperature stress. Interestingly, the *gsnor1* function-deficient mutants significantly increased SNO levels in plant cells [60], which supports our results. However, under 50 μM cadmium stress, GSNOR activity and its transcriptional expression decreased by 31%, accompanied by a decrease in NO and SNO content in peas [61]. The different changes of SNOs and GSNOR activity in these experiments may be caused by different varieties, parts and stresses. For a deeper insight, total protein levels of *S*-nitrosylation under different treatments were detected in this experiment. Consistent with the SNO level, the bands under exogenous BR treatment had the deepest color, indicating that most proteins underwent *S*-nitrosylation under this treatment. In comparison with the LT + GSNO + BRz treatment, the nitrogenated protein bands were darker under LT + BR + cPTIO. At low temperature, the protein bands were wide and dark, while the protein bands treated by exogenous GSNO were narrow. The main reasons for this phenomenon are as follows: on one hand, it may be due to the scavenging of free radicals and improvement of antioxidant capacity in response to low-temperature stress in mini Chinese cabbage seedlings, which produce large amounts of reducing agents such as glutathione and ascorbate in vivo. However, *S*-nitrosylation is not a simple enzymatic process. The S-NO bond is easily reduced by intracellular reductants, rendering it useless [62]. On the other hand, proteins may contain multiple cysteines, and because of the unstable nature of SNO, *S*-nitrosylated cysteines may be difficult to detect and distinguish from non-*S*-nitrosylated amino acids [62]. The above results implied that BR might act as an upstream signal of NO, inducing the increase of NO content in vivo and increasing the level of SNO in the process of improving cold tolerance of mini Chinese cabbage seedlings.

## 4. Materials and Methods

### 4.1. Plant Material and Growth Conditions

Uniform seeds (*Brassica rapa* ssp. *Pekinensis* var. ‘Huanai’) were soaked for 2 h and transferred to a Petri dish lined with two layers of moist filter paper for germination in an artificial climate chamber (RDN-400E-4; Southeast Instruments Co., Ltd., Ningbo, Zhejiang, China) at 28 °C in the dark. When seed germination exceeded 80%, uniformly grown seedlings were colonized on sponges, using hydroponics, for further cultivation at 26/18 °C (day/night) in an artificial climate chamber, 250 µmol m^−2^ s ^−1^, photoperiod 14/10 h, relative humidity 75%. The 1/8 Hoagland nutrient solution was renewed when the seedlings spread their first leaves, renewed every two days. After that, we increased the concentration of nutrient solution according to the seedling’s growth to renew the nutrient solution.

### 4.2. Treatments and Experimental Design

#### 4.2.1. BR Concentration Screening Experiment

In this experiment, to find the optimal BR concentration that alleviates the low-temperature stress in mini Chinese cabbage seedlings, the five-leaf stage seedlings grown in 1/2 Hoagland nutrient solution were subjected to low temperature treatment (4/4 °C, day/night). Meanwhile, BR was sprayed exogenously under the low-temperature treatment with six concentration gradients (0, 0.01, 0.05, 0.1, 0.5 and 1.0 mg/L), once a day for three days, and the low-temperature treatment for 7 days.

#### 4.2.2. NO Concentration Screening Experiment

In this experiment, to find the optimal concentration of GSNO for alleviating low-temperature stress in seedlings, five-leaf stage seedlings grown in 1/2 Hoagland nutrient solution were subjected to low-temperature treatment (4/4 °C, day/night). Meanwhile, GSNO was sprayed exogenously under the low-temperature treatment with five concentration gradients (0, 10, 50, 100 and 150 µM), once a day for three days, and the low-temperature treatment for seven days.

#### 4.2.3. S-Nitrosylation Experiment

In this experiment, based on the concentration screening results of the previous two experiments, the following treatments were set to explore the upstream or downstream relationship between BR and NO as well as the alleviating mechanism of BR on low-temperature stress: (1) normal temperature cultivation (CK); (2) low-temperature cultivation (LT); (3) low-temperature cultivation + 0.05 mg/L BR (LT + BR); (4) low-temperature cultivation + 50 µM GSNO (LT + GSNO); (5) low-temperature cultivation + 0.05 mg/L BR + 50 µM cPTIO (LT + BR + cPTIO); (6) low-temperature cultivation + 50 µM GSNO + 10 µM BRz (LT + GSNO + BRz). Among them, cPTIO (Carboxy-PTIO) is a NO scavenger and BRz (Brassinazole) is a BR biosynthesis inhibitor. During low-temperature treatment, cPTIO and BRz were sprayed first and then BR and GSNO were sprayed correspondingly after 5 h, once a day for three days, and during the low-temperature treatment for seven days. The following indexes were determined after treating for 7 d. The samples were frozen in liquid nitrogen and stored in a cryogenic refrigerator at −80 °C for determination of the relevant indexes. Each treatment was repeated three times.

### 4.3. Growth Indexes

After 7 d of treatment, the stem diameters (diameter at the base of the stem) of the seedlings were measured with vernier callipers [38]. The fresh weights of seedlings were determined using an electronic balance. Then, after drying in an oven at 105 °C for 15 min, the temperature was lowered to 80 °C and the seedlings continued to dry to a constant weight to measure the dry weight [63]. The third functional leaf from outside to inside was selected and scanned in the scanner (YMJ-C, Zhejiang Top Co., Ltd., Zhejiang, China).

### 4.4. Physiological Indexes

#### 4.4.1. Chlorophyll Content

The content of chlorophyll was determined by acetone extraction. An amount of 0.1 g of fresh leaf samples and 10 mL of 80% acetone were immersed in test tubes with a cover and placed in dark conditions for 48 h and shaken every 12 h. After 48 h, the OD values were determined at 645 nm and 663 nm, respectively.

#### 4.4.2. Malondialdehyde (MDA) Content

The content of MDA was determined by thiobarbituric acid (TBA) colorimetry: an amount of 0.5 g of plant leaves was ground with 5 mL 10% trichloroacetic acid (TCA) and then centrifuged at 3000 r/min for 10 min, with the supernatant as the sample extract. An amount of 2 mL of supernatant and 2 mL 0.5% TBA were bathed in boiling water for 20 min and centrifuged after cooling. OD values of the supernatant were measured at 532 nm, 600 nm and 450 nm, respectively [64].

#### 4.4.3. Free Proline Content

Determination of free proline by acid ninhydrin method: an amount of 0.2 g leaves was weighed, 10 mL 3% sulfosalicylic acid was added and the leaves were bathed in boiling water for 10 min. After cooling, the leaves were filtered, and the filtrate was the extract. Then, 2 mL supernatant + 2 mL H_2_O + 2 mL glacial acetic acid + 4 mL 2.5% acidic ninhydrin were bathed in boiling water for 30 min. After cooling, the supernatant liquid was taken to measure the OD value at 520 nm [65].

### 4.5. S-Nitrosylation Level Index

#### 4.5.1. NO Content Determination

NO content was determined using a kit (Suzhou Keming Biotechnology Co., Ltd., Keming, Suzhou, China). The specific methods and steps were carried out according to the kit instructions.

#### 4.5.2. SNO Content Determination

The content of SNO was determined by the Saville-Gress method referring to Puyaubert et al. [66] with minor revisions. After grinding 0.5 g leaves into powder with liquid nitrogen, we added 600 μL extraction buffer containing 1 mM PMSF, incubated on ice for 20 min and centrifuged at 10,000× *g* at 4 °C for 15 min. We added 50 μL supernatant to 1% sulfa with or without 0.2% (*w*/*v*) HgCl_2_ and incubated in darkness for 20 min (without control). Then, we added 100 μL 0.02% NED [N-(1-naphthalene)-ethylenediamine] and incubated for 5 min. The absorbance at 540 nm was measured by an enzyme standard instrument (SpectraMax Absorbance Reader, Molecular Devices, San Jose, CA, USA).

#### 4.5.3. Quantitative Real-Time PCR (qRT-PCR) of BrGSNOR

We extracted total RNA with a Plant RNA Extraction Kit (Tiangen, Beijing, China). Reverse transcription was performed with a *PrimeScript^TM^ RT reagent* Kit (Tiangen, Beijing, China). Expression of *BrGSNOR* gene was determined by qRT-PCR with primer F: TCTCTGTCACACCGACGCCTAC; R: TTACACCTTCGCCAACACTCTCAAC. *BrA**CTIN* was used as an internal reference and the primer was F: CCAGGAATCGCTGACCGTAT; R: CTGTTGGAAAGTGCTGAGGGA. The qRT-PCR was performed by using Real-Time PCR System (*ABI 7500*, *THERMO FISHER*, America). The relative expressions of the *BrGSNOR* gene were computed with the 2^−∆∆Ct^ method [67].

#### 4.5.4. Determination of S-Nitrosoglutathione Reductase (GSNOR) Activity

The activity of *S*-nitroso glutathione reductase (GSNOR) was determined by Durner et al. [68]. The absorbance value at 340 nm was measured by a spectrophotometer every 30 s with 4 values, and its activity was expressed by nmol NADH consumed per milligram of protein per minute.

#### 4.5.5. Biotin Switch Assay

Total nitrosylated protein levels were determined by the biotin switch method. The extracted protein was added to the 6 × SDS loading buffer in proportion, and the centrifuge tube was sealed with a sealing film. The centrifuge tube was placed in a 100 °C water bath for 10 min and then cooled with ice. At the same time, we prepared 12% separation glue, poured it into the glass plate and sealed it with ddH_2_O. After the separation glue cooled, we poured out excess water and poured in the thick glue. After the concentrated glue cooled, we placed the glass plate with the glue into the electrophoresis tank. We added 20 µL boiled protein samples to each well. The sample quantity at the protein marker was 5 µL. An amount of 20 µL of boiled 1 × SDS loading buffer was added to the well. The electrophoresis was set at 120 V and terminated when the bromophenol blue in the protein sample ran to the bottom of the gel. After electrophoresis, the gels were stained for 30 min according to staining solution instructions (Beyotime, Zhongshan, China, 0552-500 mL), observed after staining and photographed using a gel imaging system (Amersham Imager 600, General Electric, Fairfield, CT, USA).

### 4.6. Statistical Analysis

SPSS 20.0 statistical software was used for significance analysis. Each treatment was repeated 3 times, and the data shown were mean ± SE. According to Duncan’s test, the difference was statistically significant at *p* < 0.05 and was plotted with Graph Pad Prism 8.4.0. The relative gray-scale values were analyzed by image J.

## 5. Conclusions

In conclusion, 0.05 mg/L BR and 50 μM GSNO can effectively alleviate the damage of low-temperature stress on mini Chinese cabbage. Under low-temperature stress, exogenous spraying of 0.05 mg/L BR significantly increased endogenous NO content and *S*-nitrosylation level. Similarly, the levels of *S*-nitrosylation were significantly increased when spraying BR after removing NO. Thus, 0.05 mg/L BR and 50 μM GSNO play an important role in the resistance to low-temperature stress, and BR may act as the upstream signal of NO-induced protein *S*-nitrosation to enhance the tolerance of low-temperature stress in mini Chinese cabbage.

## Figures and Tables

**Figure 1 ijms-23-10964-f001:**
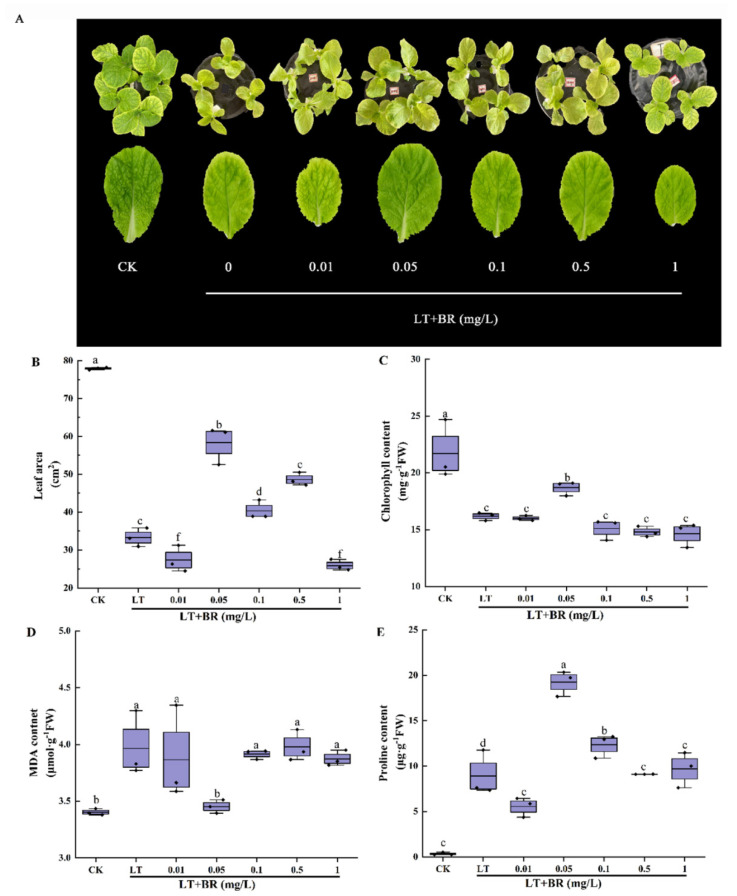
Effects of different concentrations of BR on the phenotype, leaf area, chlorophyll content, MDA and proline content of mini Chinese cabbage under low-temperature stress. Five-leaf seedlings were treated with CK (26/18 °C, day/night), LT (4/4 °C, day/night) and different concentrations of BR (0, 0.01, 0.05, 0.1, 0.5 and 1.0 mg/L) under low-temperature conditions. The leaves were sprayed with BR once a day for 3 d and treated for 7 d. After 7 days, leaf area, chlorophyll content, MDA and proline contents were measured, respectively. (**A**) Phenotype and leaf size of the third leaf from outside to inside after treating for 7 d; (**B**) Leaf area of the third leaf from outside to inside; (**C**) Chlorophyll content; (**D**) MDA content; (**E**) Proline content. Different lowercase letters above the error bars indicate significant differences at the 0.05 level (*p* ≤ 0.05).

**Figure 2 ijms-23-10964-f002:**
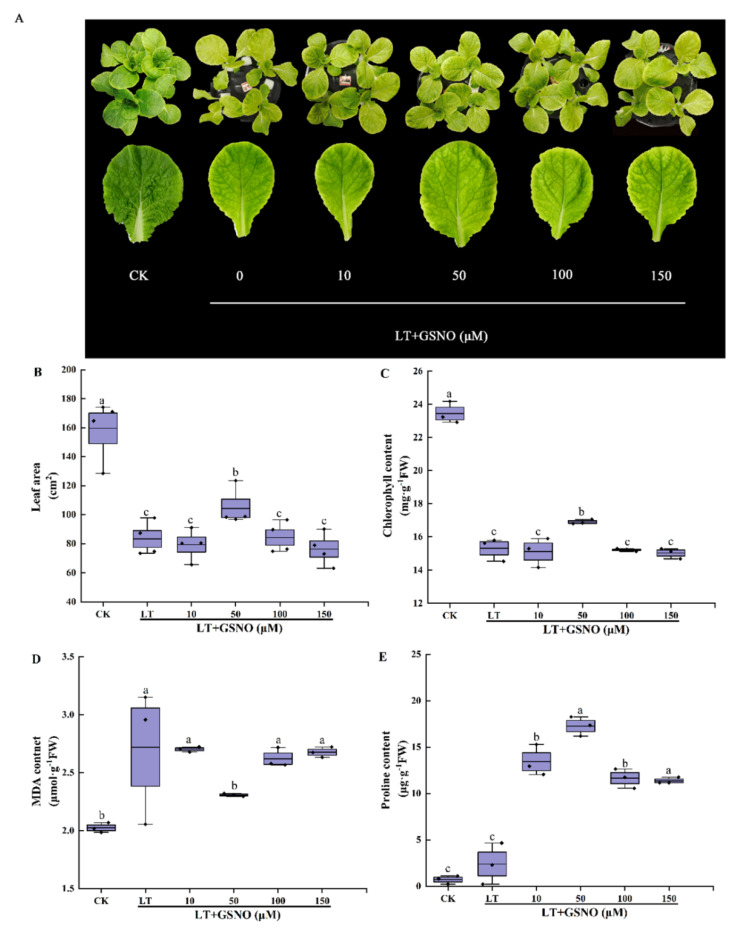
The effects of different concentrations of GSNO on the phenotype, leaf area, chlorophyll content, MDA and proline content of mini Chinese cabbage under low temperature stress. Five-leaf seedlings were treated with CK (26/18 °C, day/night), LT (4/4 °C, day/night) and different concentrations of GSNO (0, 10, 50, 100 and 150 μM) under low-temperature stress. The leaves were sprayed with GSNO once a day for 3 d and treated for 7 d. After 7 d, leaf area, chlorophyll content, MDA and proline contents were measured, respectively. (**A**) Phenotype and leaf size of the third leaf from outside to inside after treating for 7 d; (**B**) Leaf area of the third leaf from outside to inside; (**C**) Chlorophyll content; (**D**) MDA content; (**E**) Proline content. Different lowercase letters above the error bars indicate significant differences at the 0.05 level (*p* ≤ 0.05).

**Figure 3 ijms-23-10964-f003:**
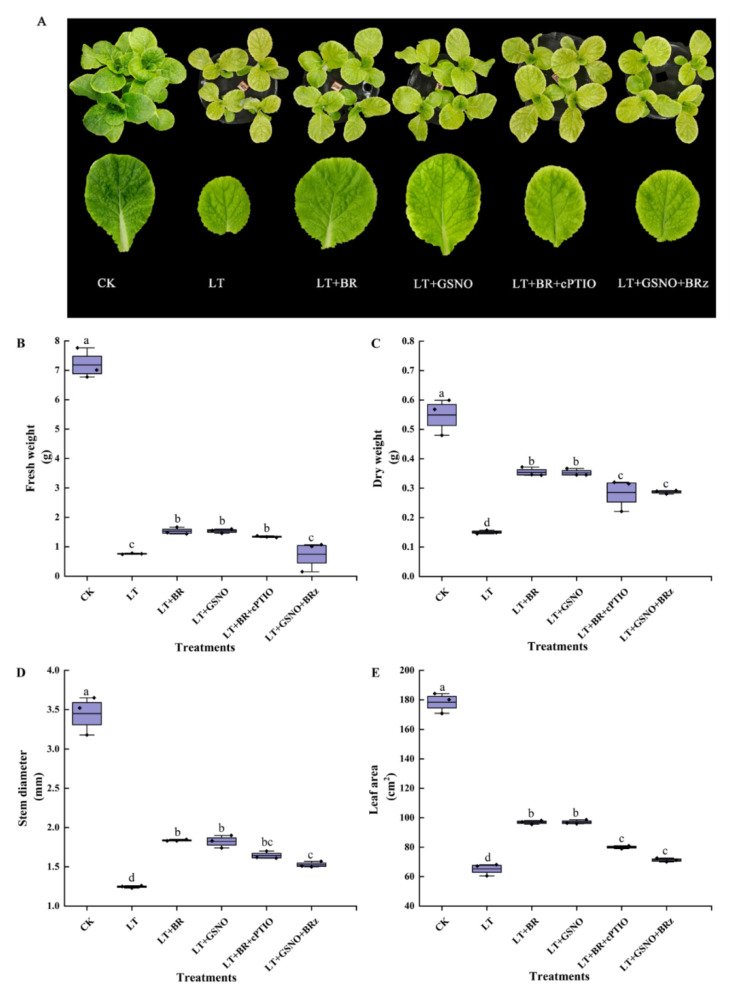
Effects of BR, GSNO, cPTIO and BRz treatments on fresh weight, dry weight, stem diameter and leaf area of mini Chinese cabbage under low temperature stress. Five-leaf seedlings were treated with CK (26/18 °C, day/night), LT (4/4 °C, day/night), LT + 0.05 mg/L BR, LT + 50 μM GSNO, LT + 0.05 mg/L BR + 50 μM cPTIO and LT + 50 μM GSNO + 10 μM BRz, cPTIO and BRz were sprayed separately, and then BR and NO were sprayed after an interval of 5 h, once every day for 3 d and treated for 7 days. After treating for 7 d, fresh weight, dry weight, stem diameter and leaf area were measured, respectively. (**A**) Phenotype and leaf size of the third leaf from outside to inside after treating for 7 d; (**B**) Fresh weight; (**C**) Dry weight; (**D**) Stem diameter; (**E**) Leaf area. Different lowercase letters above the error bars indicate significant differences at the 0.05 level (*p* ≤ 0.05).

**Figure 4 ijms-23-10964-f004:**
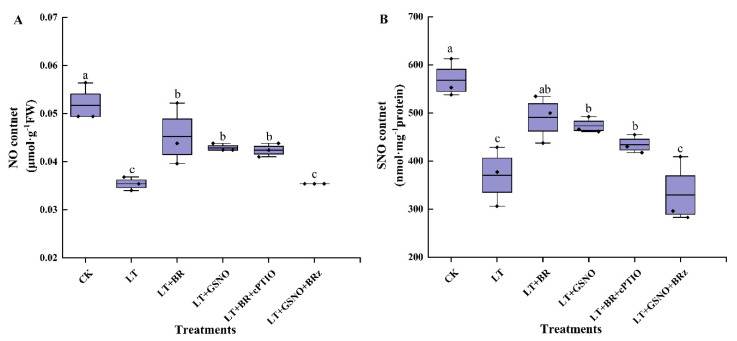
Effects of BR, GSNO, cPTIO and BRz treatments on NO content and SNO content of mini Chinese cabbage under low-temperature stress. Five-leaf seedlings were treated with CK (26/18 °C, day/night), LT (4/4 °C, day/night), LT + 0.05 mg/L BR, LT + 50 μM GSNO, LT + 0.05 mg/L BR + 50 μM cPTIO and LT + 50 μM GSNO + 10 μM BRz, cPTIO and BRz were sprayed separately, and then BR and NO were sprayed after an interval of 5 h, once every day for 3 d and treated for 7 d. After treating for 7 d, NO and SNO contents were measured, respectively. (**A**) NO content. (**B**) SNO content. Different lowercase letters above the error bars indicate significant differences at the 0.05 level (*p* ≤ 0.05).

**Figure 5 ijms-23-10964-f005:**
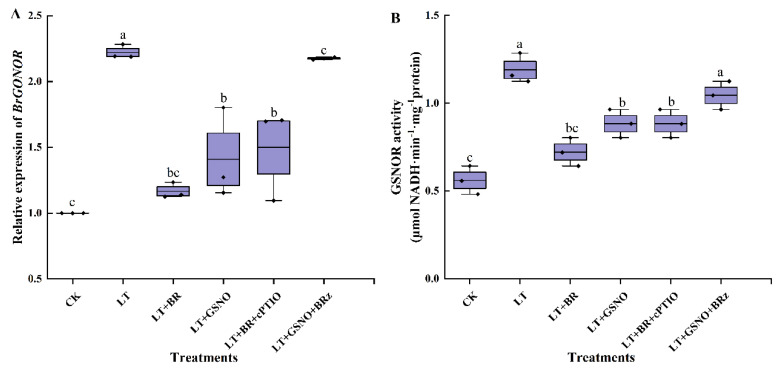
Effects of BR, GSNO, cPTIO and BRz treatments on *BrGSNOR* gene relative expression levels and GSNOR activity of mini Chinese cabbage under low temperature stress. Five-leaf seedlings were treated with CK (26/18 °C, day/night), LT (4/4 °C, day/night), LT + 0.05 mg/L BR, LT + 50 μM GSNO, LT + 0.05 mg/L BR + 50 μM cPTIO and LT + 50 μM GSNO + 10 μM BRz. After 7 d, relative expression of *BrGSNOR* gene and GSNOR activity were measured, respectively. (**A**) relative expression level of *BrGSNOR* gene; (**B**) GSNOR enzyme activity. Different lowercase letters indicate significant differences at the 0.05 level.

**Figure 6 ijms-23-10964-f006:**
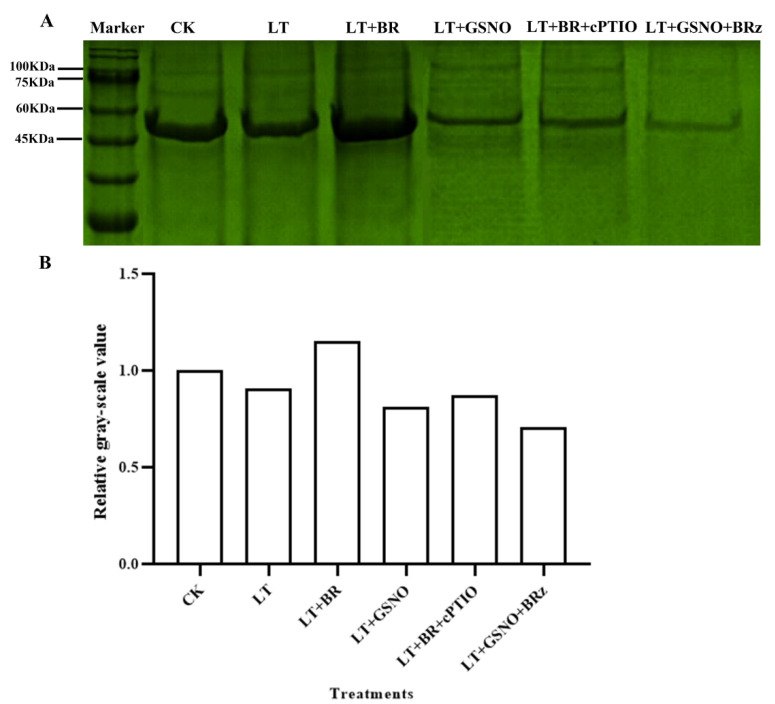
Effects of BR, GSNO, cPTIO and BRz treatments on total nitrosylated protein in Chinese cabbage under low-temperature stress. Total nitrosylated protein was determined by biotin switch assay. (**A**) Staining results of total nitrosylated protein in mini Chinese cabbage under CK (26/18 °C, day/night), LT (4/4 °C, day/night), LT + 0.05 mg/L BR, LT + 50 μM GSNO, LT + 0.05 mg/L BR + 50 μM cPTIO and LT + 50 μM GSNO + 10 μM BRz. (**B**) The relative gray-scale value of total *S*-nitrosylated protein under different treatments.

## Data Availability

Not applicable.

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
