# Peer review of "BR-Mediated Protein *S*-Nitrosylation Alleviated Low-Temperature Stress in Mini Chinese Cabbage (*Brassica rapa* ssp. *pekinensis*)"

_ijms, 2022, doi:10.3390/ijms231810964_

Round 1

Reviewer 1 Report

1- I would recommend editing the language of the manuscript

2- The graphs and error bars (in all the figures) are not nicely seen so the authors have to mount properly their graphs using a professional program like Graph Pad Prism, also they have to change the way they indicate significance on the graph and explain that clearly in the methods section

3- The authors have to quantify total nitrosylated protein in Figure 6 and show this quantification in a graph relative to a proper control

Author Response

Dear reviewer,

            Thank you for your helpful suggestions! After carefully reading these suggestions, we have modified the manuscript accordingly, and detailed corrections are listed below point by point. Now list them as following:

Response to Reviewer 1 Comments

Point 1: I would recommend editing the language of the manuscript.

Response 1: Thanks for the reviewer's comments. We have edited the language in the revised manuscript.

Point 2: The graphs and error bars (in all the figures) are not nicely seen so the authors have to mount properly their graphs using a professional program like Graph Pad Prism, also they have to change the way they indicate significance on the graph and explain that clearly in the methods section.

Response 2: Thanks for the reviewer's comments. With your comments, we have redrawn figures with professional software Graph Pad Prism 8.4.0 and we made corresponding revisions in the materials and methods: “According to Duncan's test, the difference was statistically significant at P < 0.05 and was plotted with Graph Pad Prism 8.4.0. The relative gray-scale values were analyzed by image J” (Line 537-538). Error bars (in all the figures) were redrawn. The way we indicated significance is Duncan's test. The Dunnett's rule is applicable to the comparison between the means of multiple experimental groups and the means of control groups. Similarly, many references also use this method to express the significance between treatments. For example:

Fang, P.; Yan, M.; Chi, C.; Wang, M.; Zhou, Y.H.; Zhou, J.; Shi, K.; Xia, X.; Foyer, C.H.; Yu, J. Brassinosteroids Act as a Positive Regulator of Photoprotection in Response to Chilling Stress. Plant Physiol. 2019, 180, 2061-2076. https://doi.org/10.1104/pp.19.00088

Heidari, P.; Entazari, M.; Ebrahimi, A.; Ahmadizadeh, M.; Vannozzi, A.; Palumbo, F.; Barcaccia, G.J.H. Exogenous EBR Ameliorates Endo genous Hormone Contents in Tomato Species under Low-Temperature Stress. Hort. 2021, 7, 4.

Point 3: The authors have to quantify total nitrosylated protein in Figure 6 and show this quantification in a graph relative to a proper control.

Response 3: Thanks for the reviewer's comments. With your comments, we have quantified nitrosylated proteins under different treatments by imageJ. As shown in Figure 6B, the relative gray-scale value of nitrosylated proteins under different treatments. (Line 289)

Thank you again for your useful comments and suggestions on the improvement of our manuscript.

The manuscript has been resubmitted to your journal. We look forward to your positive response.

Yours sincerely,

Linli Hu

Reviewer 2 Report

The manuscript "BR mediated protein S-nitrosylation alleviated low temperature stress in mini Chinese cabbage (Brassica rapa ssp. pekinensis)" by Xueqin Gao, Jizhong Ma, Jianzhong Tie, Yutong Li, Linli Hu and Jihua Yu is a clear little experimental work.

The manuscript presents qualitative data allowing one to make an assumption about one of the concomitant mechanisms of the action of sinosteroids. The work may be accepted for publication with minor changes.

So, it is necessary to clarify and highlight the purpose of the work ... verification of the assumption ... at the end of the introduction.

Secondly, it is required to clarify why this temperature and this duration of exposure were chosen. It remains unclear to me the use of the word stress, which is a specific concept and has not been confirmed by the authors in any way, that this is stress, and not just an adverse effect.

It is also somewhat surprising that the discussion of the results considers only the NO content option without considering other possible explanations.

Author Response

Response to Reviewer 2 Comments

Point 1: So, it is necessary to clarify and highlight the purpose of the work ... verification of the assumption ... at the end of the introduction.

Response 1: Thanks for the reviewer's comments. At the end of the introduction part of the first manuscript, we have clarified and highlight the purpose of the work: “So far, studies on the role of BR in plant response to abiotic stresses have been increasing gradually, but few reports have focused on BR involvement in NO-induced S-nitrosylation to alleviate the low temperature stress of mini Chinese cabbage(Line 120-123). Meanwhile, we also described the verification of the assumption on this work: “ In order to explore whether exogenous BR alleviates low temperature stress in mini Chinese cabbage seedlings by increasing NO content and enhancing S-nitrosylation level of proteins, in this study, we first cleared that exogenous BR and GSNO at optimum concentration could alleviate low temperature stress in mini Chinese cabbage, and the upstream or downstream relationship of BR and NO was discussed, finally the protein S-nitrosylation level induced by BR and GSNO under low temperature stress were measured. Our results preliminarily demonstrated that BR acts as an upstream signal of NO, induced protein S-nitrosylation which was involved in alleviating low temperature stress in mini Chinese cabbage seedlings(Line 123-130). Moreover, the following experimental plan based on the experimental results were put forward: “The specific S-nitrosylation proteins induced by BR under low temperature would be identified in the following experiment, to further elaborate the mechanism of BR-mediated protein S-nitrosylation modification to improve the low temperature tolerance of mini-Chinese cabbage, which provided a new strategy for the management of mini Chinese cabbage under low temperature and a new research idea and direction for the mitigation mechanism of BR on low temperature stress(Line 130-135).

Point 2: Secondly, it is required to clarify why this temperature and this duration of exposure were chosen. It remains unclear to me the use of the word stress, which is a specific concept and has not been confirmed by the authors in any way, that this is stress, and not just an adverse effect.

Response 2: Thanks for the reviewer's comments. First, we finally chose 4°C as the cold stress temperature for mini Chinese cabbage in this experiment based on a review of the references on low temperature stress. Second, given the duration of low temperature stress at 4°C. We set two time periods in the preliminary experiment, 3d and 7d, respectively. We found no significant changes in the phenotype of mini Chinese cabbage treated with 3d of low temperature compared with the phenotype of normal temperature, which did not cause cold stress to mini Chinese cabbage. In contrast, after 7d of low temperature treatment, the leaves of mini Chinese cabbage were yellowed, veins were clearly visible, and the leaf edges lost water and wilted. Thus, this experiment finally set 7d as the time of cold stress.

References on low temperature stress:

Lee, J., Lim, YP., Han, CT. et al. Genome-wide expression profiles of contrasting inbred lines of Chinese cabbage, Chiifu and Kenshin, under temperature stress. Genes Genom2013, 35, 273–288. https://doi.org/10.1007/s13258-013-0088-2

Dai, Y., Zhang, L., Sun, X. et al. Transcriptome analysis reveals anthocyanin regulation in Chinese cabbage (Brassica rapa L.) at low temperatures. Sci. Rep.  2022, 12, 6308. https://doi.org/10.1038/s41598-022-10106-1

Fang, P.; Yan, M.; Chi, C.; Wang, M.; Zhou, Y.H.; Zhou, J.; Shi, K.; Xia, X.; Foyer, C.H.; Yu, J. Brassinosteroids Act as a Positive Regulator of Photoprotection in Response to Chilling Stress. Plant Physiol. 2019, 180, 2061-2076. https://doi.org/10.1104/pp.19.00088

Arfan, M.; Zhang, D.-W.; Zou, L.-J.; Luo, S.-S.; Tan, W.-R.; Zhu, T.; Lin, H.-H. Hydrogen Peroxide and Nitric Oxide Crosstalk Mediates Brassinosteroids Induced Cold Stress Tolerance in Medicago truncatulaInt. J. Mol. Sci. 201920, 144. https://doi.org/10.3390/ijms20010144

The phenotype of seedlings treated with low temperature for 3 d and 7 d respectively.

Point 3: It is also somewhat surprising that the discussion of the results considers only the NO content option without considering other possible explanations.

Response: Thanks for the reviewer's comments. Because the argument of this experiment is that BR alleviates low temperature stress in mini Chinese cabbage by increasing the level of nitrosylation in vivo, and the occurrence of protein S-nitrosylation is based on the level of endogenous NO, which is the process of redox reaction between NO and sulfhydryl groups on cysteine residues to generate SNO. Therefore, in the discussion section, we focus on the levels of NO and SNO, as well as on the denitrosylation level (the expression of GSNOR and its enzymatic activity). Moreover, in the discussion, we discuss not only the NO content but also the possibility that it is due to the instability of SNO when we analyze the reasons for the narrow band of S-nitrosylated proteins after exogenous spraying of GSNO: “At low temperature, the protein band were wide and dark, while the protein band treated by exogenous GSNO were narrow. The main reasons for this phenomenon are as follows: on one hand, it may be due to the scavenging of free radicals and improvement of antioxidant capacity in response to low-temperature stress in mini Chinese cabbage seedlings, which produce large amounts of reducing agents such as glutathione and ascorbate in vivo. However, S-nitrosylation is not a simple enzymatic process. the S-NO bond is easily reduced by intracellular reductants, rendering it useless [62]. On the other hand, proteins may contain multiple cysteines, and the unstable nature of SNO, S -nitrosylated cysteines may be difficult to detect and distinguish from non-S -nitrosylated amino acids [62]. The above results implied that BR might act as an upstream signal of NO, inducing the increase of NO content in vivo and increasing the level of SNO in the process of improving cold tolerance of mini Chinese cabbage seedlings” (Line 421-431).

Thank you again for your useful comments and suggestions on the improvement of our manuscript.

The manuscript has been resubmitted to your journal. We look forward to your positive response.

Yours sincerely,

Linli Hu

Reviewer 3 Report

The submitted MS focused on exogenous application of Brassinosteroid (BR) and NO donor salt in alleviating adverse effects of low temperature stress on mini cabbage. In addition, the authors focused on possible interaction or cross talk between BR signaling and NO mediated signaling through S-nitrosylation. The idea is nice. However, the authors are unable to establish the probable cross talk between the BR signaling and NO-mediation S-nitrosylation. It is suggested to write briefly BR signaling component start from BR receptor protein including BZR1/BES1  important transcription factors that mediate cold resistance in plants. Cite some references in which BR signaling activated S-nitrosylation. Then describe NO mediated signaling for cold stress resistance. Establish the objectives of the study. 
Abstract: Various abbreviations are used it needs to be explained.
Introduction: Line 56 ... what is meant by apoplectic ...? I think it is apoplastic ...
Line 65-67 .... Needs revisions
Therefore, exogenous BR plays an important regulatory role in alleviating low temperature stress in plants, but the mechanism of how BR alleviates low temperature stress in mini Chinese cabbage has not been reported yet.
Discussion: 
First paragraph needs to be revised 
Conclusion: It also needs to be revised. 

Author Response

Response to Reviewer 3 Comments

Point 1: The submitted MS focused on exogenous application of Brassinosteroid (BR) and NO donor salt in alleviating adverse effects of low temperature stress on mini cabbage. In addition, the authors focused on possible interaction or cross talk between BR signaling and NO mediated signaling through S-nitrosylation. The idea is nice. However, the authors are unable to establish the probable cross talk between the BR signaling and NO-mediation S-nitrosylation. It is suggested to write briefly BR signaling component start from BR receptor protein including BZR1/BES1 important transcription factors that mediate cold resistance in plants. Cite some references in which BR signaling activated S-nitrosylation. Then describe NO mediated signaling for cold stress resistance. Establish the objectives of the study. 
Response 1: Thanks for the reviewer's comments. These opinions are of great significance to the research purpose of this paper. We have revised the introduction in the revised manuscript. The details are as follows:

(1) BR signaling component and BR receptor protein including BZR1/BES1 that mediate cold resistance in plants. We have made a revision in the introduction at lines 52 to 64: “In the presence of BRs, the kinase activity of BRI1 is activated, and BRI1 interacts with its co-receptor BAK1 (BRI1-associated receptor kinase 1) to form a complex. BRI1 and BAK1 phosphorylate each other and release the BRI1 inhibitor BKI1 (BRI1 kinase inhibitor 1) at the same time. Then the phosphate cascade process is completed through BSK1 (BR-signaling kinase 1), CDG1 (Constitutive differential growth 1) and other signalling elements, which activates BSU1 (BRI1-suppressor 1) to inhibit BIN2 and dephosphorylates BES1/BZR1 [6]. Interestingly, BZR1 (Brassinazole resistant 1), an important transcription factor for BR signalling, can bind directly to the promoter regions of CBF1 and CBF2 (CBF/DREB) and promote their expression in response to low temperature stress, further illustrating the promotion of BR signalling for low temperature resistance [7]. Meanwhile, BR induces BZR1 to directly activate the expression of the RBOH1 (respiratory burst oxidase homolog 1) and the accumulation of apoplastic H2O2, thereby increasing the protein involved in photoprotection, suggesting that this process may be related to the autophagic pathway [8].

(2) NO mediated S-nitrosylation signaling for cold stress resistance:We have made a revision in the introduction at lines 89 to 98:Abat et al.[18] reported that 17 different S-nitrosylation sites were identified in low temperature stressed Brassica juncea. In addition, the content of SNO under low temperature stress was significantly increased by 1.4 times at 1 h. Low temperature affects the photosynthesis of plants and reduces their photosynthetic efficiency of plants [19]. It was found that the activity of GSNOR was significantly increased in peas induced by low temperature. At the same time, the contents of NO and SNO also increased, which caused S-nitrosylation in pea to relieve the low temperature stress [20]. Moreover, forty-eight S-nitrosylated proteins were identified in the germ extraneous of mustard seedlings. S-nitrosylation increased the activities of dehydroascorbate reductase (DHAR ) and glutathione S-transferase (GST ) to remove reactive oxygen species, thereby enhancing cold tolerance [21].

(3) BR signaling activated NO on cold stress resistance and the purpose of our study: We have made a revision in the introduction at lines 115 to 128: “Ding et al. [27] showed that in cold-stressed cucumber, NO content increased after 24-epibrassinolide (EBR) treatment and decreased after inhibiting BR biosynthesis. Moreover, young alfalfa (Medicago truncatula) leaves treated at 4°C were sprayed with BR and BRz (BR biosynthesis inhibitor), which a large amount of NO was accumulated under BR treatment. However, when BR biosynthesis was inhibited, the content of NO decreased significantly and reduced cold resistance [28]. So far, studies on the role of BR in plant response to abiotic stresses have been increasing gradually, but few reports have focused on BR involvement in NO-induced S-nitrosylation to alleviate the low temperature stress of mini Chinese cabbage. To explore whether exogenous BR alleviates low temperature stress in mini Chinese cabbage seedlings by increasing NO content and enhancing the S-nitrosylation level of proteins, in this study, we first cleared that exogenous BR and GSNO at optimum concentration could alleviate low temperature stress in mini Chinese cabbage, and the upstream or downstream relationship of BR and NO was discussed, finally, the protein S-nitrosylation level induced by BR and GSNO under low temperature stress were measured.

Point 2: Abstract: Various abbreviations are used it needs to be explained.
Response 2: Thanks for the reviewer's comments. We have added the full names of the relevant abbreviations to the abstract in response to comments. Line 19-20: S-nitrosoglutathione (GSNO), Line 21: malondialdehyde (MDA), Line 24: S-nitrosothiol (SNO), Line 27: S-nitrosoglutathione reductase (GSNOR).

Point 3: Introduction: Line 56 ... what is meant by apoplectic ...? I think it is apoplastic ...
Response 3: Thanks for the reviewer's comments. We were careless, we have changed “apoplectic” to “apoplastic” according to the comments. (Line 62)

Point 4: Line 65-67 .... Needs revisions. Therefore, exogenous BR plays an important regulatory role in alleviating low temperature stress in plants, but the mechanism of how BR alleviates low temperature stress in mini Chinese cabbage has not been reported yet.
Response 4: Thanks for the reviewer's comments. We have revised this sentence. The details are as follows: Therefore, exogenous BR plays an important regulatory role in alleviating low temperature stress in plants, but the mechanism of how BR alleviates low temperature stress in mini Chinese cabbage is still unclear. (Line:70-72)

Point 5: Discussion: First paragraph needs to be revised 
Response 5: Thanks for the reviewer's comments. We have revised the first paragraph of discussion. The details are as follows:

Cold and frost damage are common environmental factors which many crops face in the northwest of China, severely limiting crop production [26]. Low temperatures lead to reduced seed germination, significant inhibition of plant growth and reproduction, and substantial reduction in crop yield and quality [27,28]. Early studies have shown that BR played important functions in cold stress [29,30]. Exogenous BRs enhanced the expression of CBF1 and COR47 after low temperature treatment, indicating that BRs enhanced the expression and low temperature resistance of Arabidopsis CBF [31]. In addition, NO is an active nitrogen species that can directly participate in plant response to abiotic stress through S-nitrosation modification. S-nitrosation of the transcription factor LeCBF1 was found in tomato seedlings treated with cold damage [32]. In this study, low temperature inhibited the growth and chlorophyll synthesis of the seedlings, resulting in yellow leaves. At the same time, the content of MDA increases, aggravating the degree of membrane lipid peroxidation, thus damaging the integrity of cell membrane. The content of free proline is also affected, which is reduced by low temperature and leads to cell water loss. In contrast, exogenous spraying of 0.05 mg/L BR and 50 µM GSNO effectively alleviated the low-temperature damage in mini Chinese cabbage. Therefore, this study investigated the alleviating effects of exogenous BR and GSNO on low temperature and the upstream or downstream relationship between BR and NO. From the perspective of S-nitrosylation modification induced by BR, the mechanism of alleviating low temperature stress by BR was preliminarily clarified. (Line:297-315)

Point 6: Conclusion: It also needs to be revised. 

Response 6: Thanks for the reviewer's comments. We have revised the conclusion. The details are as follows:

In conclusion, 0.05 mg/L BR and 50 μM GSNO can effectively alleviate the damage of low temperature stress on mini Chinese cabbage. Under low temperature stress, exogenous spraying 0.05 mg/L BR significantly increased endogenous NO content and S-nitrosylation level. Similarly, the levels of S-nitrosylation were significantly increased when spraying BR after removing NO. Thus, 0.05 mg/L BR and 50 μM GSNO play an important role in the resistance to low temperature stress, and BR may act as the upstream signal of NO-induced protein S-nitrosation to enhance the tolerance of low temperature stress in mini Chinese cabbage. (Line 540-546)

Thank you again for your useful comments and suggestions on the improvement of our manuscript.

The manuscript has been resubmitted to your journal. We look forward to your positive response.

Yours sincerely,

Linli Hu

Round 2

Reviewer 1 Report

I think the authors were able to cover most of my concerns, I would accept the paper for publication.